# Genome expansion in early eukaryotes drove the transition from lateral gene transfer to meiotic sex

**Marco Colnaghi[1,2], Nick Lane[1,2], Andrew Pomiankowski[1,2]***

[1]CoMPLEX, University College London, London, United Kingdom; [2]Department of Genetics, Evolution and Environment University College London, London, United Kingdom

**Abstract** Prokaryotes acquire genes from the environment via lateral gene transfer (LGT). Recombination of environmental DNA can prevent the accumulation of deleterious mutations, but LGT was abandoned by the first eukaryotes in favour of sexual reproduction. Here we develop a theoretical model of a haploid population undergoing LGT which includes two new parameters, genome size and recombination length, neglected by previous theoretical models. The greater complexity of eukaryotes is linked with larger genomes and we demonstrate that the benefit of LGT declines rapidly with genome size. The degeneration of larger genomes can only be resisted by increases in recombination length, to the same order as genome size – as occurs in meiosis. Our results can explain the strong selective pressure towards the evolution of sexual cell fusion and reciprocal recombination during early eukaryotic evolution – the origin of meiotic sex.

## Introduction

Understanding the origin and maintenance of sex in the face of multiple costs was long considered the 'Queen of problems in evolutionary biology' (*Bell, 1982*). Sexual reproduction breaks up advantageous combinations of alleles, halves the number of genes transmitted to the offspring, and is less efficient and energetically more costly than asexual reproduction (*Bell, 1982*; *Otto and Lenormand, 2002*; *Otto, 2009*). In spite of these disadvantages sex is a universal feature of eukaryotic life. The presence of common molecular machinery, widespread among all eukaryotic lineages, is a strong indication that the Last Eukaryotic Common Ancestor (LECA) was already a fully sexual organism (*Schurko and Logsdon, 2008*; *Speijer et al., 2015*). Meiotic genes are commonly found in putative asexual eukaryotes, including Amoebozoa (*Lahr et al., 2011*; *Hofstatter et al., 2018*), Diplomonads (*Ramesh et al., 2005*), Choanoflagellates (*Carr et al., 2010*) and even lineages once thought to be early diverging such as *Trichonomas vaginalis* (*Malik et al., 2008*). Eukaryotic asexuality is not ancestral but a secondarily evolved state. The selective pressures that gave rise to the origin of meiotic sex must therefore be understood in the context of early eukaryotic evolution.

Phylogenomic analysis shows that eukaryotes arose from the endosymbiosis between an archaeal host and a bacterial endosymbiont, the ancestor of mitochondria (*Müller et al., 2012*; *Williams et al., 2013*; *Martin et al., 2015*; *Zaremba-Niedzwiedzka et al., 2017*). The presence of energy-producing endosymbionts allowed the first eukaryotes to escape the bioenergetic constraints that limit the genome size and cellular complexity of prokaryotes (*Lane and Martin, 2010*; *Lane, 2020*). Extra energetic availability came with the evolutionary challenge of the coexistence of two different genomes within the same organism. As with other endosymbioses, the symbiont genome underwent a massive reduction, with the loss of many redundant gene functions (*Timmis et al., 2004*; *López-Madrigal and Gil, 2017*). Alongside this, symbiont release of DNA into the host's cytosol caused the repeated transfer of genes to the host genome, many of which were

**\*For correspondence:**
a.pomiankowski@ucl.ac.uk

**Competing interests:** The authors declare that no competing interests exist.

retained, contributing to the massive genome size expansion during early eukaryotic evolution (*Timmis et al., 2004*; *Martin and Koonin, 2006*; *Lane, 2011*).

Both the host and the endosymbiont, like modern archaea and proteobacteria, are likely to have been capable of transformation – the uptake of exogenous DNA from the environment followed by homologous recombination (*Bernstein and Bernstein, 2013*; *Vos et al., 2015*; *Ambur et al., 2016*). This process involves the acquisition of foreign DNA, the recognition of homologous sequences and recombination, and therefore presents striking similarities with meiosis in eukaryotes. The *Rad51/Dcm1* gene family, which plays a central role in meiosis, has high protein sequence similarity with *RecA*, which promotes homologous search and recombination in prokaryotes (*Lin et al., 2006*; *Johnston et al., 2014*). It has been suggested that *RecA* was acquired by the archaeal ancestor of eukaryotes via endosymbiosis from its bacterial endosymbiont (*Lin et al., 2006*). Alternatively, the *Rad51/Dcm1* family could have evolved from archaeal homologs of *RadA* (*Seitz et al., 1998*). Regardless, the presence of this common molecular machinery and the striking similarities between these processes suggest that meiosis evolved from bacterial transformation (*Schurko and Logsdon, 2008*; *Bernstein and Bernstein, 2013*; *Mirzaghaderi and Hörandl, 2016*). But the selective pressures that determined this transition are still poorly understood.

Historically, the main focus of the literature on the origin and the maintenance of sex has been the comparison of sexual and clonal populations, or the spread of modifiers that increase the frequency of recombination (*Bell, 1982*; *Otto, 2009*). Recombination can eliminate the linkage between beneficial and deleterious alleles due to Hill-Robertson effects (*Felsenstein and Yokoyama, 1976*; *Barton and Otto, 2005*), increase adaptability in rapidly changing (*Hamilton, 1980*; *Gandon and Otto, 2007*; *Jokela et al., 2009*) or spatially heterogeneous (*Pylkov et al., 1998*; *Lenormand and Otto, 2000*) environments, and prevent the accumulation of deleterious mutations due to drift predicted by Muller's ratchet for asexual populations (*Muller, 1964*; *Haigh, 1978*). These benefits of recombination outweigh the multiple costs of sexual reproduction and explain the rarity of asexual eukaryotes (*Otto, 2009*). But remarkably, they provide us with virtually no understanding of why bacterial transformation was abandoned in favour of reciprocal meiotic recombination. The real question is not why sex is better than clonal reproduction, but why did meiotic sex evolve from prokaryotic transformation?

Lateral Gene Transfer (LGT) has been recognised as a major force shaping prokaryotic genomes (*Ochman et al., 2000*; *Lapierre and Gogarten, 2009*; *Vos et al., 2015*). Unlike conjugation and transduction, mediated respectively by plasmids and phages, transformation is the only LGT mechanism to be exclusively encoded by genes present on prokaryotic chromosomes (*Ambur et al., 2016*), and is maintained by natural selection because it provides benefits analogous to those of sexual recombination (*Vos, 2009*; *Levin and Cornejo, 2009*; *Wylie et al., 2010*; *Takeuchi et al., 2014*; *Vos et al., 2015*). Recombination via transformation allows adaptation by breaking down disadvantageous combinations of alleles (*Levin and Cornejo, 2009*; *Wylie et al., 2010*) and preventing the fixation of deleterious mutations (*Levin and Cornejo, 2009*; *Takeuchi et al., 2014*). Some theoretical studies (*Redfield, 1988*; *Redfield et al., 1997*) suggest that transformation is only advantageous in the presence of strong positive epistasis, a condition rarely met by extant prokaryotes. But more recent modelling work shows that transformation facilitates the elimination of deleterious mutations and prevents Muller's ratchet (*Levin and Cornejo, 2009*; *Takeuchi et al., 2014*). As transformation provides similar advantages to meiotic sex, why did the first eukaryotes forsake one for the other? How did the unique conditions at the origin of eukaryotic life give rise to the selective pressures that determined this transition? In particular, is it possible that the massive expansion in genome size in early eukaryotes created the conditions for the evolution of a more systematic way of achieving recombination?

We know very little about the relation between genome size and the accumulation of deleterious mutations in populations undergoing transformation, as previous models either do not consider it explicitly (*Levin and Cornejo, 2009*; *Wylie et al., 2010*) or treat it as a constant parameter (*Takeuchi et al., 2014*). In order to evaluate the impact of genome size and recombination rate on the dynamics of accumulation of mutation, we develop a new theoretical and computational model of Muller's ratchet in a population of haploid individuals undergoing homologous recombination of genetic material via transformation. For simplicity, we refer to this process simply as 'LGT'. We do not consider genetic exchange facilitated by plasmids or other selfish genetic elements (*Ochman et al., 2000*; *Lapierre and Gogarten, 2009*; *Vos et al., 2015*).

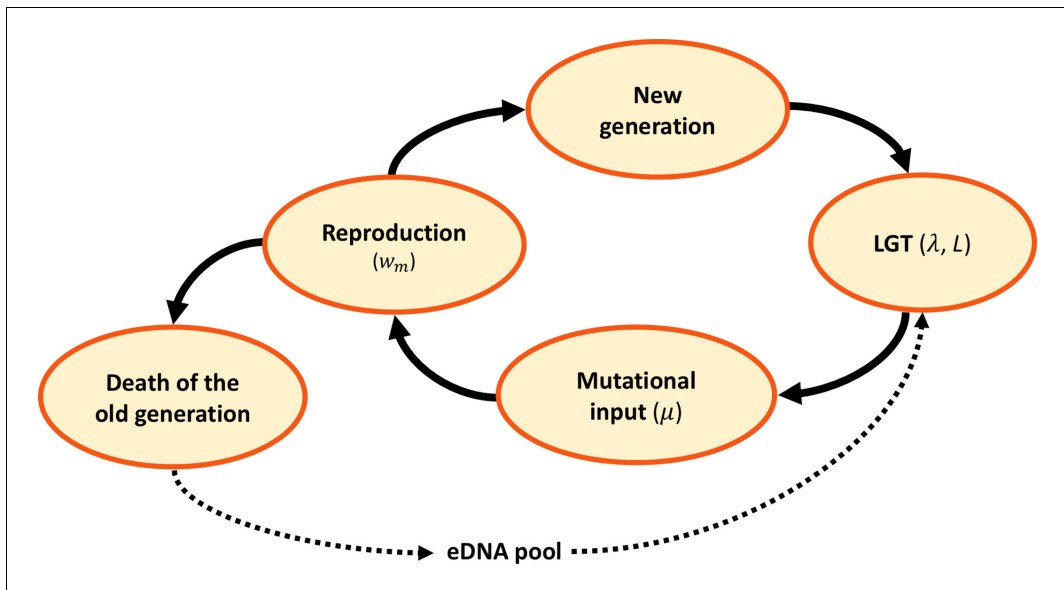

**Figure 1.** Model dynamics. After the birth of a new generation, eDNA is acquired from the environment and randomly recombined with recombination length $L$, at a rate $\lambda$ per genome. Following LGT, mutations are randomly introduced at a rate $\mu$ per locus. A new generation is then sampled at random, in proportion to reproductive fitness $w_m$. The old generation dies and its DNA is released, constituting the eDNA pool for the new generation.

The population is subjected to variable rates of mutation and recombination due to LGT. Our model includes two new parameters, genome size and recombination length, which have not been taken into account by previous theoretical studies (*Levin and Cornejo, 2009*; *Wylie et al., 2010*; *Takeuchi et al., 2014*). We evaluate the effects of different selective landscapes, either uniform across the genome, or split between core and accessory genomes. The severity of the ratchet is evaluated using standard approaches for measuring the rate of fixation of deleterious mutations and the expected extinction time of the fittest class (*Haigh, 1978*; *Gordo and Charlesworth, 2000*; *Takeuchi et al., 2014*). We suggest that systematic recombination across the entire bacterial genomes was a necessary development to preserve the integrity of the larger genomes that arose with the emergence of eukaryotes, giving a compelling explanation for the origin of meiotic sex.

## Results

We use a Fisher-Wright process with discrete generations to model the evolution of haploid organisms with genome size $g$, and population size $N$ (*Figure 1*). They undergo homologous recombination via LGT at a rate $\lambda$ of recombination length $L$ (*Figure 1*). The environmental DNA (eDNA) available for recombination is a random portion of the genomes of the previous generation. After undergoing LGT, each genome acquires deleterious mutations at a rate $\mu$ per gene. Then, reproduction takes place, using a multiplicative fitness function which depends on the number of mutations (see Materials and Methods for more details).

### Constant mutation rate per locus

Genome size increases the severity of the ratchet, measured by $T_{ext}$, the expected extinction time of the least loaded class, LLC (*Figure 2*). Large genomes gain *de novo* mutations at a faster rate than small ones, leading to a decline in LLC extinction time, as there are more independent loci that can possibly fix for the mutant (*Figure 2*, no LGT). LGT reduces the severity of the ratchet and increases the expected LLC extinction time (*Figure 2*), making the population less vulnerable to stochastic fluctuations in population size. The beneficial effect is more evident as recombination length ($L$) and LGT rate ($\lambda$) increase (*Figure 2*). However, as genome size ($g$) increases the expected extinction time plummets, rapidly approaching that of a clonal population with a larger genome, both in the

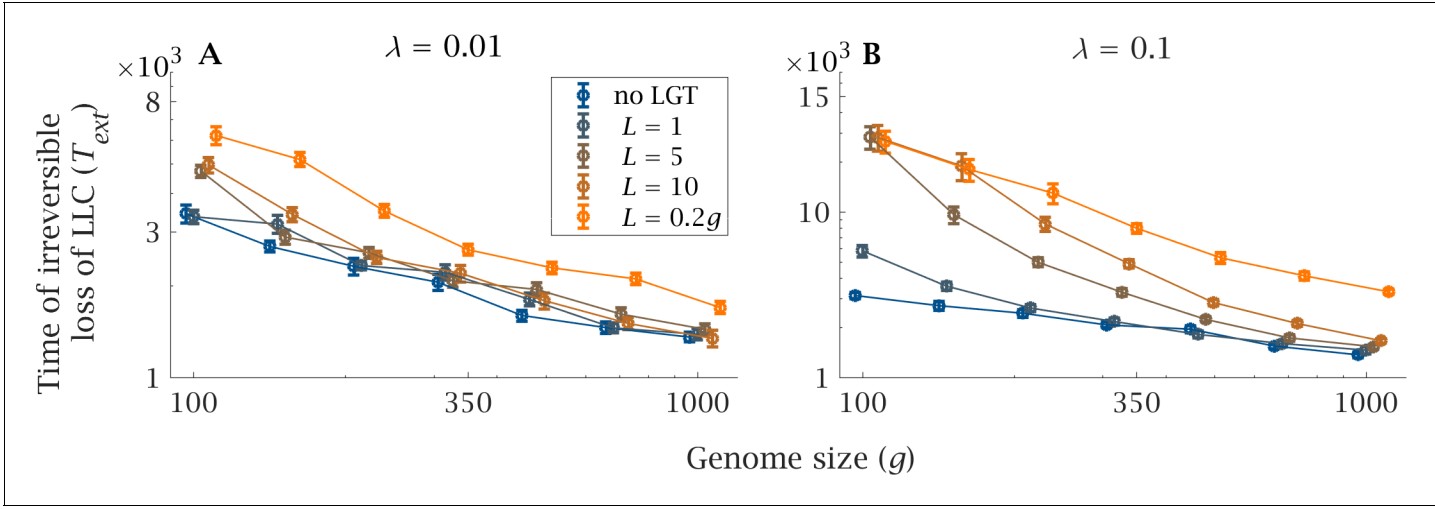

**Figure 2.** Impact of LGT and genome size on the ratchet. The mean extinction time (generations) of the Least-Loaded Class ($T_{ext}$) is shown as a function of genome size ($g$) for various recombination lengths ($L$), in the presence of (**A**) low ($\lambda = 0.01$) and (**B**) high ($\lambda = 0.1$) LGT rates. The blue lines show the extinction time when there is no LGT, and is the same in (**A**) and (**B**). Parameters: $s = 10^{-3}$, $N = 5 \times 10^3$, $\mu = 10^{-4}$, $U = \mu \times g$. Error bars show the standard deviation over 50 independent iterations.

The online version of this article includes the following source data for figure 2:

**Source data 1.** Time of extinction of least loaded class.

presence of high ($\lambda = 0.1$, ***Figure 2A***) and low ($\lambda = 0.01$, ***Figure 2B***) LGT rates. The sole exception is when recombination length is of the same order as the magnitude of genome size ($L = 0.2g$, ***Figure 2***). Only under this condition can increases in genome size be tolerated without a drastic decline in $T_{ext}$.

The rate of accumulation of deleterious mutations shows an analogous pattern (***Figure 3***). As genome size increases, the rate of mutation accumulation markedly increases, both genome wide and per locus (***Figure 3***). In a small genome, LGT reduces the speed at which mutations accumulate in a population, counteracting the ratchet effect both in the presence of a high ($\lambda = 0.1$) or low ($\lambda = 0.01$) LGT rate (***Figure 3***). This effect is more pronounced with higher LGT rates ($\lambda$) and longer recombination lengths ($L$). But even in presence of LGT, large genomes are subjected to higher rates of accumulation, comparable to those of a purely clonal population (***Figure 3***). Only when recombination length approaches the same order of magnitude as genome size ($L = 0.2g$) and occurs at high frequency ($\lambda = 0.1$) can LGT sufficiently repress mutation accumulation in large genomes (***Figures 3B and D***).

## Constant genome-wide mutation rate

The limitations of LGT in large genomes become evident in simulations with a constant genome-wide mutation rate in which we increase genome size but constrain the genome-wide mutation rate ($U$) to a constant value. In this case, changes in the efficacy of LGT cannot be due to changes in the rate at which new mutations arise in a genome.

With a low LGT rate ($\lambda = 0.01$), recombination of short fragments ($L = 1$) is not enough to halt the mutational advance and provides virtually no benefits compared to clonal reproduction, with either a low (***Figure 4A***) or high genome-wide mutation rate (***Figure 4B***). With a high LGT rate ($\lambda = 0.1$, ***Figure 4B and D***), larger values of $L$ (i.e. $L \geq 5$) can drastically reduce the rate of accumulation of mutations in small genomes, but are not able to counter Muller's ratchet in larger genomes. As in the previous section, this effect is less evident with a low LGT rate (***Figures 4A and C***). However, when the recombination length approaches the same order of magnitude as genome size ($L = 0.2g$), the accumulation of mutation load is retarded regardless of genome size, both in the presence of a low ($U = 0.01$, ***Figures 4A and B***) and high genome-wide mutation rates ($U = 0.1$, ***Figures 4C and D***). In the presence of a high LGT rate ($\lambda = 0.1$), recombination of such long fragments ($L = 0.2g$)

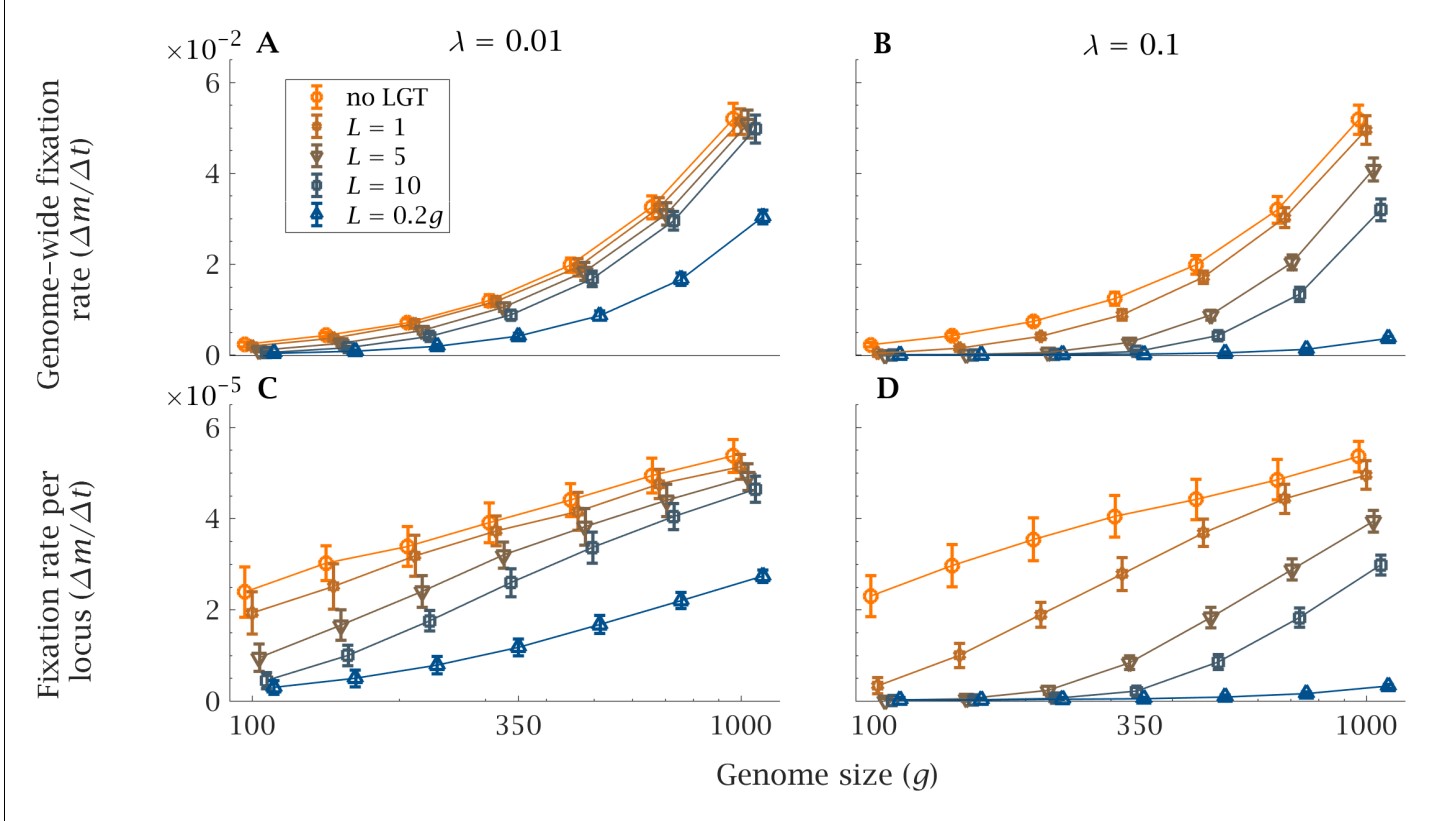

**Figure 3.** Impact of LGT and genome size on the rate of accumulation of mutation. The mean genome-wide rate of fixation of deleterious mutations per generation, calculated over a time interval $t = 10^5$ generations, as a function of genome size ($g$) for various recombination lengths ($L$). This is shown for (A) low ($\lambda = 0.01$) and (B) high ($\lambda = 0.1$) LGT rates. Similarly, the rate of fixation of deleterious mutation per locus per generation is shown, again for (C) low ($\lambda = 0.01$) and (D) high ($\lambda = 0.1$) LGT rates. As genome size increases, LGT becomes less effective in reducing the mutational burden of a population. An increase in recombination length improves the efficiency of LGT in preventing the accumulation of mutations, but this beneficial effect declines rapidly with genome size. Only if recombination length is of the same order of magnitude as genome size ($L = 0.2g$) and the rate of LGT is high ($\lambda = 10^{-1}$) can large genomes be maintained in a mutation-free state. Parameters: $s = 10^{-3}$, $N = 10^4$, $\mu = 10^{-4}$, $U = \mu \times g$. Error bars show the standard deviation over 50 independent iterations.

The online version of this article includes the following source data for figure 3:

**Source data 1.** Rate of accumulation of mutations with variable genome-wide mutation rate.

effectively prevents any increase in mutation load (*Figure 4C and D*). We even see a decline in mutation accumulation as genome size increases (*Figure 4D*). But this is for the artefactual reason that for a fixed genome wide mutation rate, the per gene mutation rate declines with $g$. This results in a net benefit as genome size increases under the condition that $L = 0.2g$ where the recombination length remains a constant fraction of $g$. Even though unrealistic, constraining the genome-wide mutation rate reveals that LGT becomes less efficient *per se* as genome size increases.

## Non-uniform strength of selection

Different loci in the genome are typically under different strengths of selection. In order to capture this fact in our model, we consider the core and accessory genomes differently. The size of the core genome is fixed ($g_c = 50$), while the accessory genome size increases as the genome expands. The core loci are under strong selection ($s = 0.005$) and the accessory loci are under weak selection ($s = 0.001$). Core and accessory loci are randomly distributed in the genome.

Under this selection regime, mutations preferentially accumulate in the accessory genome, where the strength of selection is lower, while the core genome accumulates mutations at a relatively slow rate (*Figure 5*). Genome size expansion results in more severe ratchet effects, with a marked

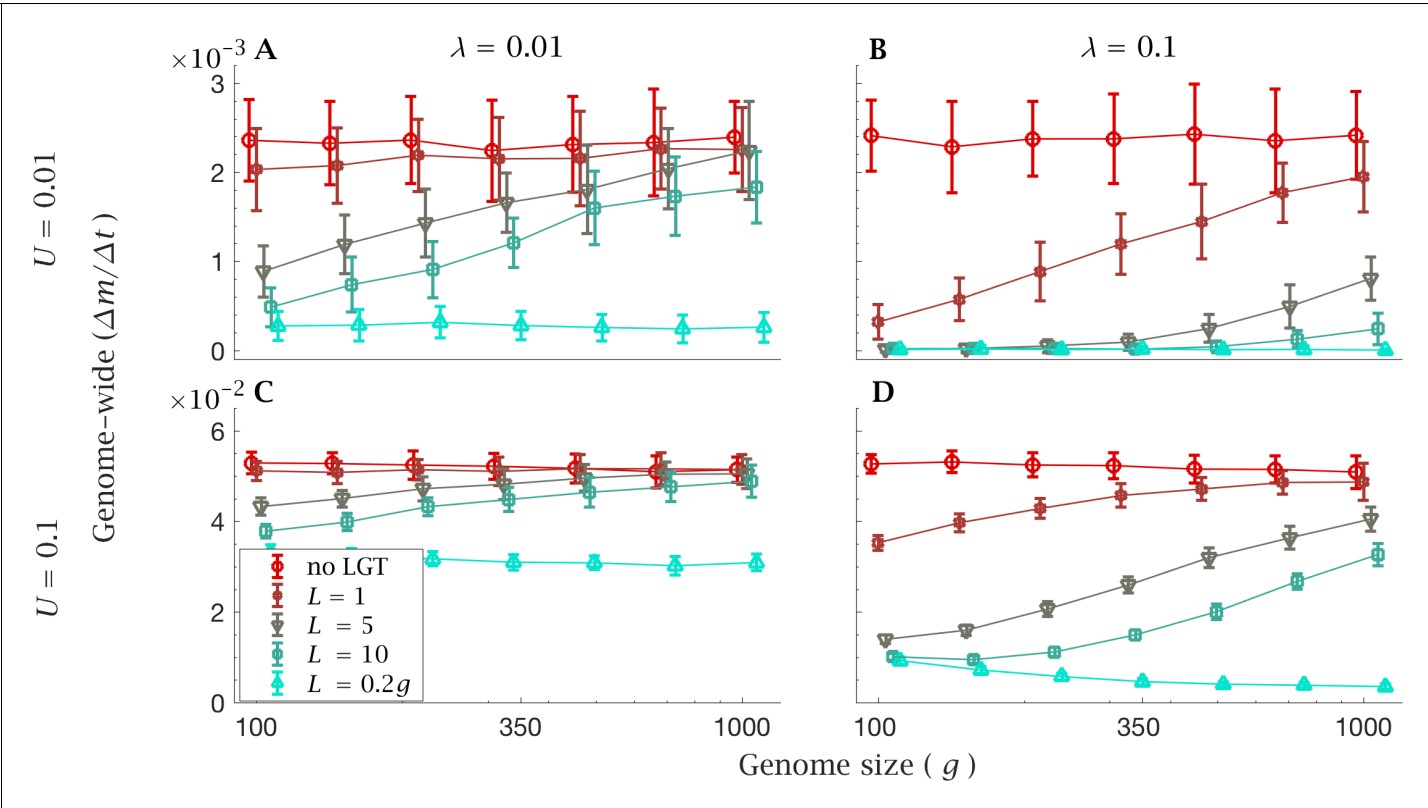

**Figure 4.** Rate of fixation of mutations with constant genome-wide mutation rate. The mean genome-wide rate of fixation of deleterious mutations per generation. This is shown for a low rate genome-wide mutation rate $U = 0.01$ for (A) low ($\lambda = 0.01$) and (B) high ($\lambda = 0.1$) LGT rates, and a high genome-wide mutation rate $U = 0.1$ for (C) low ($\lambda = 0.01$) and (D) high ($\lambda = 0.1$) LGT rates, across a range of recombination lengths (L). The fixation rate was calculated over a time interval of $t = 10^5$ generations, as a function of genome size. LGT is not able to prevent the accumulation of mutation in large genomes, except for a recombination length on the order of the whole genome ($L = 0.2g$). Parameters: $s = 10^{-3}$, $N = 5 \times 10^3$. Error bars show the standard deviation over 50 independent iterations. The different data points have been slightly off-set in order to prevent overlap between errorbars.

The online version of this article includes the following source data for figure 4:

**Source data 1.** Rate of fixation of mutations with constant genome-wide mutation rate.

increase in the rate of mutations reaching fixation in the regions of the genome that are under weaker selection, alongside a moderate increase in core genome mutation fixation rate (*Figure 6*). LGT is effective in reducing the mutational burden, both in the accessory and in the core genome; but this beneficial effect is less evident in large genomes than in small ones (*Figure 6*). Recombination across the whole genome ($L = 0.2g$) completely eliminates fixation in the core genome, regardless of genome size, and markedly reduces the fixation rate in the accessory genome, facilitating genomic expansion (*Figure 6*).

## Discussion

Asexual organisms are well known to be vulnerable to the effects of drift, which reduces the genetic variation within a population, causing the progressive and inescapable accumulation of deleterious mutations known as Muller's ratchet (*Muller, 1964*; *Haigh, 1978*; *Otto, 2009*). In eukaryotes, sexual recombination counters the effects of genetic drift and restores genetic variance, increasing the effectiveness of purifying selection and preventing mutational meltdown (*Otto, 2009*). In prokaryotes, sexual fusion does not occur. But the exchange of genetic material does occur through transformation, the lateral gene transfer (LGT) and recombination of environmental DNA (eDNA). Meiotic recombination likely arose from bacterial transformation. Understanding the reasons why this transition occurred during early eukaryotic evolution are critical to a rigorous understanding of the Queen of problems in evolutionary biology, the origin of sex. Sex did not arise from cloning, as tacitly

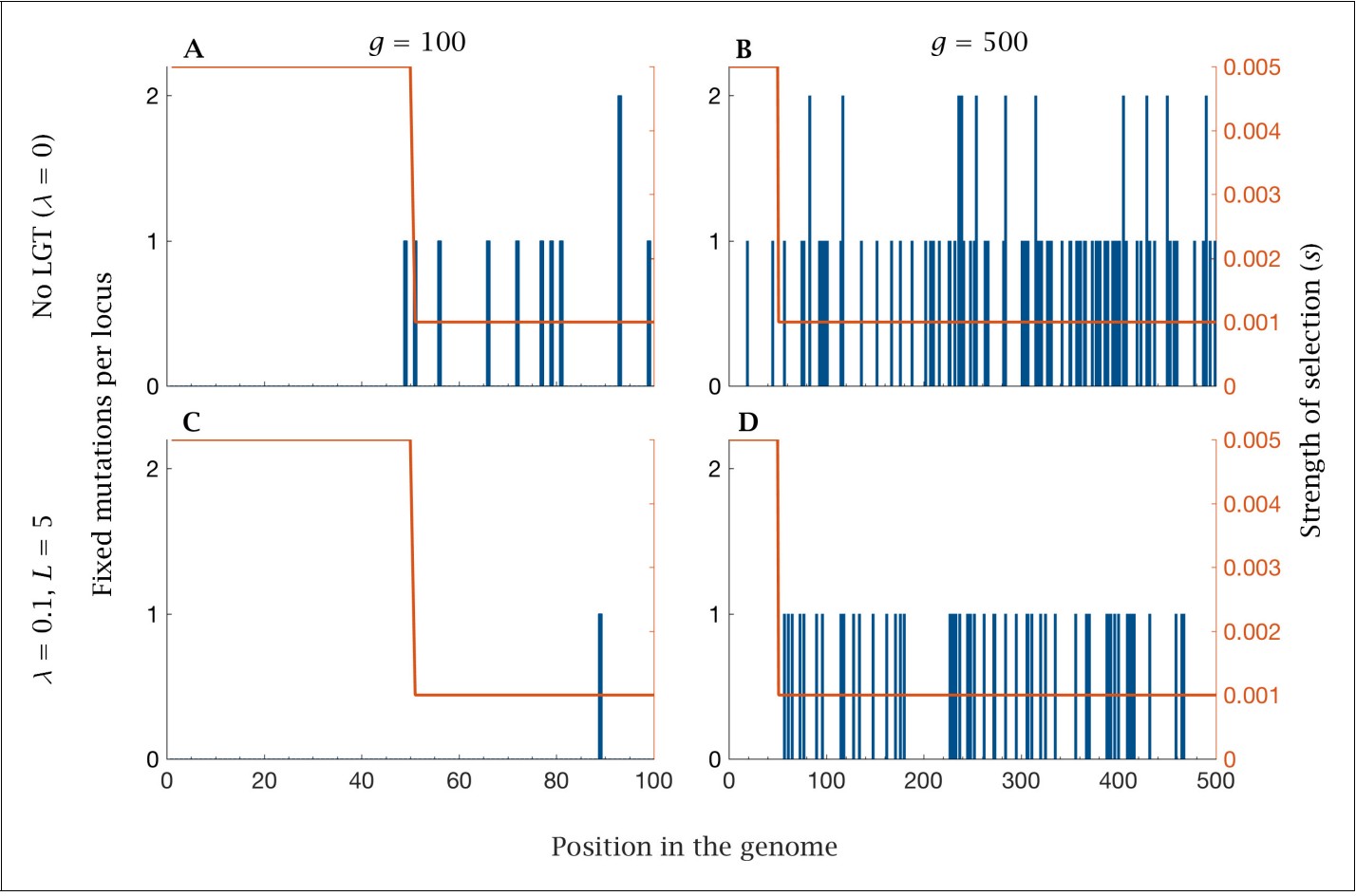

**Figure 5.** | Fixation of mutations in the core and accessory genome. Fixed mutations in the core and accessory genome after $t = 10^5$ generations for no LGT with (A) small ($g = 100$) and (B) large ($g = 500$) genome size, and for LGT ($\lambda = 0.1$, $L = 5$) with (C) small ($g = 100$) and (D) large ($g = 500$) genome size. Mutations preferentially accumulate in the accessory genome under weaker selection ($s = 0.001$), while the strongly selected core genome ($s = 0.005$) accumulates few or no mutations. The rate of fixation increases with genome size, while the benefits of LGT decline with genome size. Parameters: $N = 10^4$, $\mu = 10^{-4}$, $U = \mu \times g$.

assumed in the classic theoretical literature, but from prokaryotic transformation, a very different question which we explored. To elucidate this transition, we examined the effectiveness of LGT at countering the dynamics of Muller's ratchet, to understand where and why LGT becomes ineffective at maintaining genome integrity, necessitating the transition to sexual reproduction in early eukaryotes.

We assessed the effect of LGT on the severity of the ratchet using the expected extinction time of the least-loaded class and the rate of fixation of deleterious mutations. Unlike previous modelling work, we included genome size as a variable as opposed to a constant (e.g. 100; *Takeuchi et al., 2014*). Genome size is plainly important in relation to the evolution of eukaryotes, which have expanded considerably in almost every measure of genome size (e.g. DNA content, number of protein-coding genes, size of genes, number of gene families, regulatory DNA content; *Lane and Martin, 2010*). Considering gene number in our model reveals a strong inverse relationship between genome size and the benefits of LGT. In small genomes, LGT is effective at preventing Muller's ratchet, with long extinction times (*Figure 2*) and low rates of mutation accumulation (*Figure 4*). These results mirror those from modelling assuming a single genome size ($g = 100$), which show that LGT can halt Muller's ratchet even if the environmental DNA has a higher mutation load than the population (*Takeuchi et al., 2014*). However, our results show that large genomes limit the efficiency of LGT, increasing the overall input of mutations to the genome. Larger genomes increase the severity of the ratchet leading to shorter extinction times and faster rates of mutation accumulation

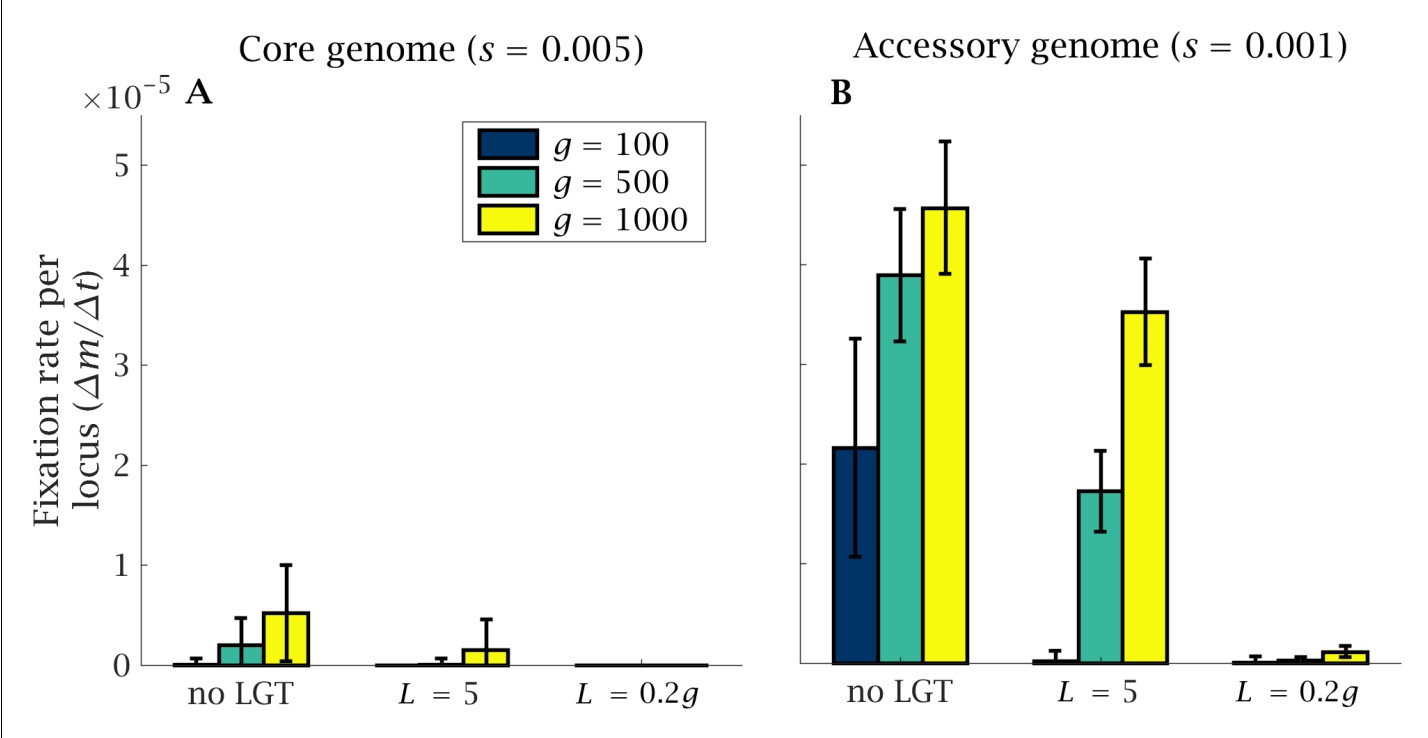

**Figure 6.** Rate of fixation of mutations in the core and accessory genome. The fixation of mutants in (**A**) the core and (**B**) the accessory genome is shown after $t = 10^5$ generations, normalised by genome size. A higher number of mutations accumulate in the accessory genome that is under weaker selection. Genome size expansion increases the severity of the ratchet and the number of fixed mutations in the core and accessory genome. The introduction of LGT considerably reduces the mutational burden. Parameters: $\lambda = 10^{-1}$, $L = 5$, $N = 10^4$, $\mu = 10^{-4}$, $U = \mu \times g$, $s_{core} = 0.005$ and $s_{acc} = 0.001$.

(*Figures 2–3*). In order to avoid unnecessary complexity, our model does not take into account other possible benefits of LGT. In particular, a number of other theoretical studies indicate that LGT can increase the rate of adaptive evolution (*Iles et al., 2003*; *Levin and Cornejo, 2009*; *Wylie et al., 2010*). In light of our results, it would be reasonable to expect these adaptive benefits decline as genome size increases.

Here we deliberately fixed genome size to evaluate the efficacy of LGT in the face of mutation accumulation. It would be illuminating to model the spread of modifiers, both of recombination length and the rate of LGT, given different variable genome size. As in the literature on organisms with meiotic sex (*Iles et al., 2003*; *Barton and Otto, 2005*), the frequency of modifiers of LGT-induced recombination will spread by hitchhiking as they generate variance in fitness, dependent on population size and the number of loci. This approach could be extended to ask how the spread of modifiers of LGT lead to changes in genome size. This could be attempted within a general model in which there is a fitness advantage of acquiring new genes through LGT balanced by the rate of deletion (*Sela et al., 2016*). The role of recombination length and the rate of LGT in genome size expansion and the frequency of gene loss needs to be clarified, and we intend to address this question in future work.

Under the assumption of a constant mutation rate per locus, theoretical results predict that the severity of the ratchet (measured as the equilibrium number of individuals in the least-loaded class) increases exponentially with genome size (*Haigh, 1978*; Appendix 1; *Appendix 1—figure 1*). In addition to this effect – which could potentially be offset by a lower mutation rate per locus – large genomes are penalized by the decreased effectiveness of LGT at reducing the mutation load. This result holds true regardless of the assumption that large genomes present a greater mutational target than smaller ones. Even with a constant genome-wide mutation rate, LGT becomes less effective at purging large genomes from deleterious mutations, showing that this effect is not due to a higher mutation rate, but to an intrinsic limitation of LGT (*Figure 4*).

The increased potency of the ratchet as genome size increases is ameliorated by an increase in the rate of LGT ($\lambda$; *Figures 2–3*). Is this a viable option for prokaryotic species to enable them to expand genome size? In a number of species, LGT has been estimated as being the same magnitude (or higher) as the rate of nucleotide substitution, including *Bacillus cereus* (*Hao and Golding, 2006*), *Streptococcus* (*Marri et al., 2006*), *Corynebacterium* (*Marri et al., 2007*), and *Pseudomonas syringae* (*Nowell et al., 2014*). Rates are highly variable among species (*Croucher et al., 2012*; *Vos et al., 2015*). Competence for transformation can be induced by a range of environmental stressors including DNA damage, high cell density and limited nutrient availability (*Bernstein and Bernstein, 2013*). But LGT rates are constrained by eDNA availability, which depends on the amount of DNA in the environment and the degree of sequence homology (*Croucher et al., 2012*; *Vos et al., 2015*). The model predicts that higher LGT rates will strengthen purifying selection and favour the elimination of mutants. This result is compatible with the strong correlation observed between the number of horizontally transferred genes and genome size across a range of prokaryotes (*Jain et al., 2003*; *Fuchsman et al., 2017*). However, it is not clear to what extent the rate of LGT can be modified. Our modelling suggests that larger bacterial genomes are more likely to be sustained by higher rates of LGT, but the benefits of LGT as actually practiced by bacteria – the non-reciprocal uptake of small pieces of DNA comprising one or a few genes – are unlikely to sustain eukaryotic-sized genomes. In short, we show that LGT as actually practised by bacteria cannot prevent the degeneration of larger genomes.

Importantly, we show that the benefits of LGT also increase with recombination length (*L*; *Figures 2–4*). In gram-positive bacteria, recombination of large eDNA sequences is the exception rather than the rule (*Croucher et al., 2012*; *Mell et al., 2014*). Experimental work indicates that the distribution of eDNA length acquired is skewed towards short fragments, with a third of transformation events less than 1kb, a median around 2-6kb and range extending up to ~50kb (*Croucher et al., 2012*). This appears to be an evolved state in *Streptococcus pneumoniae*, as the dedicated system cleaves eDNA into smaller fragments before recombination takes place (*Claverys et al., 2009*). Some studies have reported a larger median and range for transfer sizes (*Hiller et al., 2010*). Given that loci are around 1kb, with short intergenic regions, this represents the potential for several genes to be transferred in a single LGT event (*Mira et al., 2001*; *Moran, 2002*). There are several potential reasons for cells to focus on small genomic pieces in LGT recombination. Cleavage of eDNA into smaller sequences increases the likelihood of homologous recombination, while the acquisition of long sequences can be associated with loss of genetic information (*Croucher et al., 2012*) and can potentially disrupt regulatory and physiological networks (*Bernstein and Bernstein, 2013*). It has also been suggested that the small size of recombination fragments is a mechanism for preventing the spread of mobile genetic elements (*Croucher et al., 2016*). On the other hand, gram-negative bacteria do not cleave eDNA on import, but their ability to acquire eDNA sequences >50kb is limited by physical constraints (*Mell and Redfield, 2014*). The high variability of LGT size suggests that there is flexibility and the potential for evolutionary change. But there is no evidence that larger genome size is accompanied by a higher recombination length. To our knowledge bacteria do not load large pieces of chromosome via LGT (i.e. >10% of a genome), although in principle it should be possible for them to do so.

As for purely asexual populations (*Haigh, 1978*), the strength of selection plays a critical role in determining the rate of mutation accumulation, with regions of the genome under strong selection accumulating mutations at a low rate (*Figures 5–6*). The ratchet effect is mainly observed in the accessory genome, with mutations accumulating preferentially in loci under weak selection (*Figures 5–6*). Our model predicts that genome size expansion can occur in populations under strong purifying selection (e.g. due to a larger effective population size). Strong selection decreases the rate of genetic information loss, allowing the acquisition of new genetic content without an attendant increase in mutation fixation. This prediction is in agreement with the positive correlation observed between genome size and dN/dS in bacteria (*Novichkov et al., 2009*; *Bobay and Ochman, 2018*). However, organisms under similar selective pressures often display a broad range of genome sizes (*Novichkov et al., 2009*), indicating that other factors, including mutation rate and LGT, have a strong impact on prokaryotic genome size. Under high mutation rate and weak selective pressure, genome size expansion is disfavoured.

Eukaryotes, including many protists, typically possess much larger genomes than prokaryotes (*Koonin, 2009*; *Elliott and Gregory, 2015*). Eukaryotic genome size expansion was favoured by the

acquisition of an endosymbiont, which evolved into the mitochondrion. This released bioenergetic constraints on cell size and allowed the evolution of genetic and morphological complexity (*Lane and Martin, 2010*; *Lane, 2014*; *Lane, 2020*). The endosymbionts underwent gene loss, a frequently observed process in extant endosymbiotic relationships (*López-Madrigal and Gil, 2017*) and transferred multiple genes to the host, enriching the host's genome size with genes of proto-mitochondrial origin (*Timmis et al., 2004*; *Martin et al., 2015*). The reduction of energetic constraints on genome size probably also facilitated gene and even whole genome duplications, leading to several thousand new gene families in LECA (*Koonin et al., 2004*), as well as lower selective pressure for gene loss after acquisition of novel gene functions by LGT (*Szöllosi et al., 2006*). The acquisition of endosymbiotic DNA is also thought to have allowed the spread of mobile genetic elements in the host cell's genome, contributing to the increase in genome size and likely increasing the mutation rate (*Timmis et al., 2004*; *Martin and Koonin, 2006*; *Rogozin et al., 2012*). Our model explicitly considers an increase in the number of selected loci. The benefits of LGT would decline in a similar fashion if genome size expansion proceeded through the acquisition of non-selected 'junk' DNA. The presence of non-coding sequence would reduce the likelihood that a locus under selection undergoes recombination in a single LGT event (corresponding to smaller $\lambda$ in the model), thus increasing the severity of the ratchet. The same would follow if increases in genome size followed from the expansion of mobile elements within the genome, although this would introduce complexities relating to repeated sequences and ectopic recombination that require further analysis.

Such a large genome brought the first eukaryotes under the threat of mutational accumulation, creating the need for stronger purifying selection in order to keep the expanded genetic content free from mutations. Our results offer a possible explanation of why this process drove the transition from LGT to meiotic recombination at the origin of sex. In small prokaryotic genomes, LGT provides sufficient benefits to maintain genome integrity, without incurring the multiple costs associated with sexual reproduction. But LGT fails to prevent the accumulation of deleterious mutations in larger genomes, promoting the loss of genetic information and therefore constraining genome size. Our model shows that genome size expansion is only possible through a proportional increase in recombination length. We considered a recombination length $L = 0.2g$, which is equivalent to 500 genes for a species with genome size of 2,500 genes – two orders of magnitude above the average estimated eDNA length in extant bacteria (*Croucher et al., 2012*). Recombination events of this magnitude are unknown among prokaryotes, possibly because of physical constraints on eDNA acquisition. Limiting factors likely include the restricted length of eDNA, uptake kinetics and the absence of an alignment mechanism for large eDNA strands (*Thomas and Nielsen, 2005*; *Baltrus, 2013*; *Croucher et al., 2016*).

The requirement for a longer recombination length $L$ cannot be achieved by LGT, which must therefore have failed to maintain a mutation-free genome, generating a strong selective pressure towards the evolution of a new mechanism of inheritance with the loss of energetic constraints on genome size. However, this magnitude of $L$ is easily achievable via meiotic sex. The transition from LGT to meiotic sex involves the evolution of cell fusion, the transition from circular to linear chromosomes, whole-chromosome alignment and homologous recombination (*Lane, 2011*; *Goodenough and Heitman, 2014*). We have not explicitly modelled the details of this process or considered the order in which these factors arose. These aspects are crucial to our understanding of the origin of meiotic sex and will be addressed in a series of future publications. Nonetheless, our results show that eukaryotes had to increase the magnitude of recombination length beyond the limits of LGT in order to permit the expansion in genetic complexity without the attendant increase in mutational burden. Eukaryotes had to abandon LGT in order to increase recombination length and maintain a large genome. Sex was forced upon us.

## Materials and methods

We use a Fisher-Wright process with discrete generations to model the evolution of a population of $N$ haploid individuals, subject to a rate of deleterious mutation $\mu$ per locus per generation, with LGT at a rate $\lambda$. The genome of an individual $j$ is described by a state vector $\vec{z}^{(j)} = \{z_1, \ldots, z_g\}$, where $g$ is the number of loci. Each locus $i$ can accumulate a number of mutations $\{0,1,2\ldots\}$. The components $z_i^{(j)}$ are the number of deleterious mutations at the $i$-th locus of the $j$-th individual. This allows us to

keep track of the number of mutants in an individual and the distribution of mutations at each locus in the population. We define fixation of a mutant at a locus when the least-loaded class (LLC) at that locus is lost. As we neglect back-mutation, fixation of a mutant is permanent. Parameters and variables of the model are summarised in *Table 1*.

We consider two different mutational regimes: constant mutation rate per locus and constant genome-wide mutation rate. In the first scenario, the genome-wide mutation rate $U = \mu g$ is calculated as the product between the mutation rate per locus per generation ($\mu$) and the number of loci ($g$). We assume that the mutation rate per locus is constant across the whole genome. Under this assumption, the severity of the ratchet increases exponentially with genome size, and this effect can only partially be offset by an increase in population size (Appendix 1 and *Appendix 1—figure 1*). In order to study the intrinsic limitations of LGT, we also investigate the effect of a constant genome-wide mutation rate $U$ for genomes of different size. We introduce a new parameter $L$, the number of contiguous genes acquired with each LGT recombination event (i.e. the size of imported DNA), which has not been taken into account by previous theoretical studies (*Levin and Cornejo, 2009*; *Wylie et al., 2010*; *Takeuchi et al., 2014*). In order to avoid unnecessary complexity, we ignore the probability of ectopic recombination, and assume that DNA strands present in the environment (eDNA pool) are only stable for one generation before decaying irreversibly.

In the first part of this study, we assume that all mutations are mildly deleterious. Each mutation at a locus and across loci causes the same decrease in individual fitness $s$. Following previous studies of Muller's ratchet (*Haigh, 1978*; *Gordo and Charlesworth, 2000*; *Takeuchi et al., 2014*), we choose a multiplicative function to model the fitness of an individual carrying $m$ mutations, given by the formula $w_m = (1 - s)^m$ (i.e. no epistasis). In the second part of the study we investigate more complex distributions of strength of selection across the genome. In particular, we differentiate between a strongly selected core genome and accessory genome under weaker selection. Which genes belong to the core and to the accessory genome is determined by random sampling. The fitness of an individual that carries $m_i$ mutations at locus $i$ is given by $w(t) = \prod_{i=1}^{g} (1 - s_i)^{m_i}$, where $s_i = 0.005$ if locus $i$ belongs to the core genome and $s_i = 0.001$ otherwise.

Each generation, the life history of the population follows the following processes (*Figure 1*). The new generation is obtained by sampling $N$ individuals at random, with replacement, from the old population. The probability of reproduction is proportional to the individual fitness $w_m$. Each individual acquires $n^{(j)}$ new deleterious mutations, where $n^{(j)}$ is a random variable drawn from a Poisson distribution with mean $U$, which is equivalent to each locus acquiring a new mutation with probability $\mu$. The number of mutations and their position in the genome are randomly determined. Then the old generation dies, and their DNA forms the genetic pool from which the new generation acquires exogenous DNA (eDNA) for recombination. Each individual has a probability $\lambda$ of undergoing LGT. For each individual that undergoes LGT, a random donor is selected from the previous generation and $L$ contiguous loci are randomly selected from its genome. The genome is assumed to be circular, so locus $g$ is contiguous with locus 1. The corresponding components of the state vector of the recipient become equal to those of the donor. This can lead both to an increase or a decrease in the mutation load of the recipient.

Simulations are started from a population free of mutation and run for 10,000 generations, with 50 replicates for a given set of parameter values. Two measures $T_{ext}$ and $\Delta m / \Delta t$ have been used to assess the effect of the ratchet (*Haigh, 1978*; *Gordo and Charlesworth, 2000*; *Takeuchi et al., 2014*). After recombination, we calculate the number of individuals in the least-loaded class (LLC),

**Table 1.** Parameters and variables.

| | |
|---|---|
| $N$ | population size |
| $\mu$ | mutation rate per locus per generation |
| $g$ | genome size (number of loci) |
| $U$ | genome-wide mutation rate |
| $s$ | strength of selection against deleterious mutations |
| $\lambda$ | LGT rate |
| $L$ | recombination length (number of loci) |

the set of individuals whose genomes are mutation free: $\vec{z}^{(j)} = \{0, 0, \ldots, 0\}$. In populations undergoing LGT the loss of the least-loaded class is reversible, unless a mutant reaches fixation at a particular locus. At this point, the mutation-free class is irreversibly lost, and we mark this as $T_{ext}$ (extinction time of the LLC). $T_{ext}$ gives an estimate of the time that a population can remain free of mutations. The second measure is the genome-wide rate of fixation $\Delta m/\Delta t$, which is calculated as the ratio between the total number of fixed mutations at the end of the simulation, and the duration of the simulation itself (i.e. 10,000 generations). The rate of fixation per single locus is the ratio between the genome-wide rate of fixation and genome size $g$. $\Delta m/\Delta t$ is a measure of the rate of accumulation of mutations.

Simulation code is available on GitHub at https://github.com/MarcoColnaghi1990/Colnaghi-Lane-Pomiankowski-2020 (*Colnaghi, 2020*; copy archived at https://github.com/elifesciences-publications/Colnaghi-Lane-Pomiankowski-2020).

## Acknowledgements

This work very much profited from comments during review made by George Constable, Paul B Rainey and Patricia Wittkopp, for which we are very grateful. MC is supported by a CoMPLEX PhD Studentship funded by bgc3 and the Department of Genetics, Evolution and Environment. AP is supported by Engineering and Physical Sciences Research Council grants (EP/F500351/1, EP/I017909/1), and Natural Environment Research Council grant (NE/R010579/1); NL is supported by the Biotechnology and Biological Sciences Research Council (BB/S003681/1) and bgc3.

## Additional information

### Funding

| Funder | Grant reference number | Author |
| --- | --- | --- |
| Engineering and Physical Sciences Research Council | EP/F500351/1 | Andrew Pomiankowski |
| Engineering and Physical Sciences Research Council | EP/I017909/1 | Andrew Pomiankowski |
| Natural Environment Research Council | NE/R010579/1 | Andrew Pomiankowski |
| Biotechnology and Biological Sciences Research Council | BB/S003681/1 | Nick Lane |
| bg3 | | Nick Lane |

The funders provided resources to support the researchers involved in this study

### Author contributions

Marco Colnaghi, Conceptualization, Software, Formal analysis, Investigation, Methodology, Writing - original draft, Writing - review and editing; Nick Lane, Andrew Pomiankowski, Conceptualization, Formal analysis, Supervision, Investigation, Methodology, Writing - original draft, Writing - review and editing

### Author ORCIDs

Marco Colnaghi (iD) https://orcid.org/0000-0002-5641-9324
Nick Lane (iD) https://orcid.org/0000-0002-5433-3973
Andrew Pomiankowski (iD) https://orcid.org/0000-0002-5171-8755

### Decision letter and Author response

Decision letter https://doi.org/10.7554/eLife.58873.sa1
Author response https://doi.org/10.7554/eLife.58873.sa2

## Additional files

### Supplementary files

• Transparent reporting form

### Data availability

No datasets were generated or analysed during the current study. Simulation code is available on GitHub at https://github.com/MarcoColnaghi1990/Colnaghi-Lane-Pomiankowski-2020 (copy archived at https://github.com/elifesciences-publications/Colnaghi-Lane-Pomiankowski-2020).

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

## Appendix 1

In absence of LGT, previous theoretical results (*Haigh, 1978*) have shown that, at equilibrium, the number of individuals in the least-loaded class (LLC) is $n_0 = Ne^{-U/s}$. Without recombination and back-mutation, the loss of the LLC is an irreversible process – a "click" of the ratchet. The magnitude of $n_0$ determines the likelihood that the least-loaded class becomes extinct because of stochastic fluctuations (i.e. genetic drift). High values of $n_0$ increase the expected time of extinction of the LLC, whereas small values make the LLC more vulnerable to stochastic fluctuations (*Muller, 1964*; *Haigh, 1978*). Therefore $n_0$ is a good indication of the speed of the ratchet (*Haigh, 1978*). Expressing the genome wide mutation rate $U$ as $\mu \times g$ allows the equilibrium number of individuals in the LLC to be rewritten $n_0 = Ne^{-\mu g/s}$. The speed of the ratchet scales exponentially with genome size and mutation rate, and is negatively correlated with the strength of selection. Crucially, the impact of genome size is much stronger than that of population size (*Appendix 1—figure 1*). For example, a 2-fold increase in genome size can increase the speed of the ratchet by several orders of magnitude, whereas even a 10-fold reduction in population size has a rather meagre effect, except at low values (*Appendix 1—figure 1*). Therefore, any increase in genome size must be balanced by a proportional increase in strength of selection in order to avoid a drastic reduction of $n_0$.

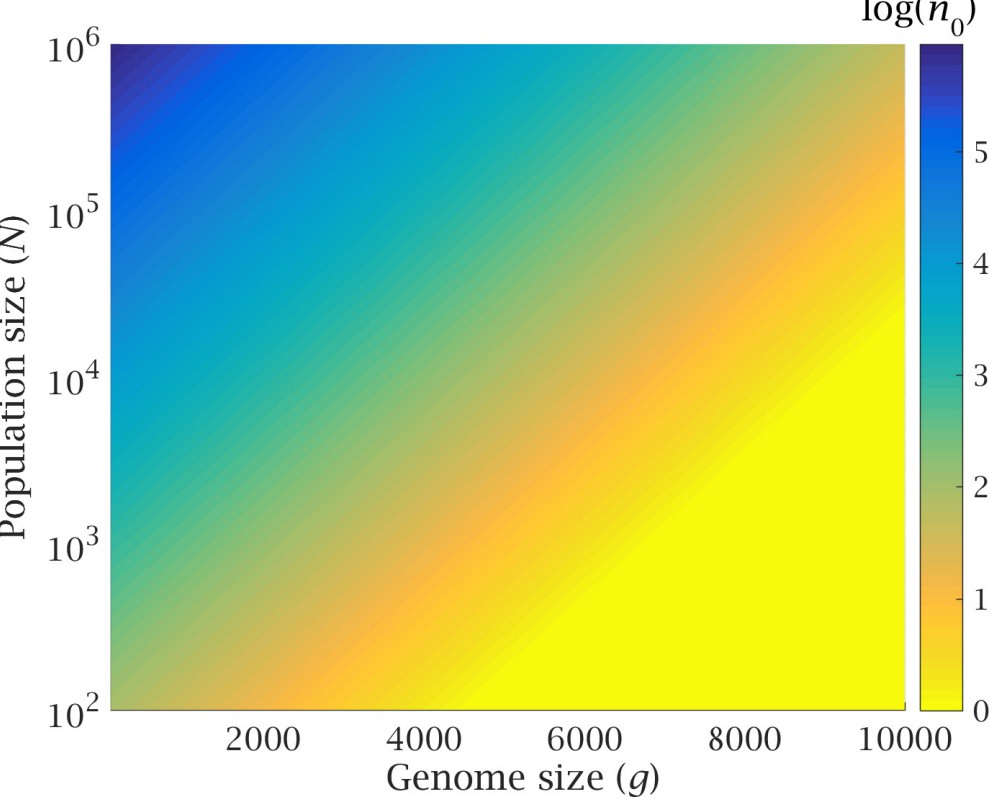

**Appendix 1—figure 1.** Genome size and population size determine $n_0$. The equilibrium number of individuals in the least-loaded class ($n_0$) is shown as a function of genome size (number of loci) and population size, with constant mutation rate $\mu = 10^{-5}$ and constant strength of selection $s = 10^{-3}$.

