## [Decision Letter]

**Acceptance summary:**

The authors investigate how the effect of Muller's ratchet is reduced by lateral gene transfer and recombination of environmental DNA, with a particular focus on how recombination length and genome size alter the magnitude of this reduction. This is a very interesting area and it's exciting to see this particular problem tackled theoretically. Both the Introduction and conclusions are strong, in particular the reorientation of the question of the paradox of sex (why did meiotic sex evolve from prokaryotic transformation?) and the focus on rapid genome expansion in early eukaryotes.

**Decision letter after peer review:**

Thank you for submitting your article "Genome expansion in early eukaryotes drove the transition from lateral gene transfer to meiotic sex" for consideration by *eLife*. Your article has been reviewed by three peer reviewers, including Paul B Rainey as the Reviewing Editor and Reviewer #1, and the evaluation has been overseen by Patricia Wittkopp as the Senior Editor. The following individual involved in review of your submission has agreed to reveal their identity: George Constable (Reviewer #2).

The reviewers have discussed the reviews with one another and the Reviewing Editor has drafted this decision to help you prepare a revised submission.

Summary:

The authors investigate how the effect of Muller's ratchet is reduced by lateral gene transfer and recombination of environmental DNA, with a particular focus on how recombination length and genome size alter the magnitude of this reduction. This is a very interesting area and it's exciting to see this particular problem tackled theoretically. Both the Introduction and conclusions are strong, in particular the reorientation of the question of the paradox of sex (why did meiotic sex evolve from prokaryotic transformation?) and the focus on rapid genome expansion in early eukaryotes.

Essential revisions:

The way the model is established, LGT provides two benefits in terms of slowing Muller's ratchet that are difficult to disentangle.

The first advantage is that, as the authors state, "Recombination via transformation allows adaptation by breaking down disadvantageous combinations of alleles". This benefit is the one that is most relevant when comparing LGT to meiotic sex, and indeed it is suggested that this is the main benefit of LGT in the paper.

The second (not evidently intentional) advantage, is that by having mutation occur after formation of the eDNA pool (see Figure 1), the eDNA pool in some sense acts like an "effective germ-line" that experiences a lower mutational load. This has the consequence of slightly lowering the effective mutation rate experienced by populations undergoing LGT, as is described below for a simplified case.

We therefore request that the authors experiment with altering the ordering of events in their model (see Figure 1) to place the mutational step before the formation of the eDNA pool. This would allow the benefits arising from transformation to be easily distinguished from the benefits potentially arising from the smaller mutational load currently experienced by the eDNA pool. The outcome stands to range between two extremes:

A) The ordering of these events has little effect on the qualitative results. The model is robust to these changes and the arguments of the paper are strengthened.

B) It may turn out that the model behaviour observed so far is entirely dependent on the reduced mutational load of the eDNA pool (rather than transformation). Although it appears unlikely, this scenario would be difficult to justify from a biological perspective and would likely undermine the value of the paper.

Issue of decreased mutational load of eDNA pool

Suppose we begin with a population with no mutations. We use this ancestral population to (1) form the eDNA pool and (2) provide the samples for the new generation. Individuals in the new generation are then subject to mutation, receiving an average number of *U* mutations each. Let's assume for simplicity that the maximum number of mutations is 1. With probability λ the individual undergoes LGT, replacing part of its genome with a sample from the mutation-free eDNA pool. With the circular genome assumed, the probability of losing a mutation acquired by an individual in the new generation during LGT is *L*/*g* (recombination length / genome length). An effective mutation rate, *U_eff_*, in this simplified scenario can then be defined

*U_eff_* = *U* (1 – λ *L*/*g*) = mu *g* ( 1 –λ *L* /*g*) [constant mutation rate per locus]

This has a number of straightforward consequences.

As *g* -> inf with *L* held constant, the effective mutation rate tends to the actual mutation rate. This *could* explain why the simulation results in Figure 2 with LGT and *L*=1-10 (which experience something like this effective mutation rate, *U_eff_*) tend to the result of the simulations without LGT (which experiences the actual mutation rate, *U*) as *g* gets large.

As *g* -> inf with *L* held proportional to *g* (i.e. *L*=0.2*g*), the effective mutation rate is always strictly lower than the actual mutation rate. This *could* explain why the simulations in Figure 2 with *L*=0.2*g* always take longer to accumulate deleterious mutations than those without LGT.

Finally, in Figure 2B the simulation results with LGT and *L*=10 are very close to those with *L*=0.2*g* when *g* is low (*g* approx100). For these parameters, we have (*L*/*g*)=0.1 and (*L*/*g*)=0.2 respectively, which can lead to similar values of *U_eff_*, and thus *could* explain the observed similarity in the average extinction time of the LLC in this parameter range.

The above discussion comes with the caveat that the numerical differences between *U* and this (admittedly crude) approximation for *U_eff_* are very low for the parameters investigated (*U_eff_* can be at most (1-λ)*U*=0.9U). Nevertheless it is important to understand which components of the model set-up are driving the dynamics. Placing mutation before the formation of the eDNA pool would circumvent the problem of these different effective mutation rates, and provide a fairer comparison of the robustness of populations under LGT and under no LGT to Muller's ratchet.

---

## [Author Response]

Essential revisions:The way the model is established, LGT provides two benefits in terms of slowing Muller's ratchet that are difficult to disentangle.The first advantage is that, as the authors state, "Recombination via transformation allows adaptation by breaking down disadvantageous combinations of alleles". This benefit is the one that is most relevant when comparing LGT to meiotic sex, and indeed it is suggested that this is the main benefit of LGT in the paper.The second (not evidently intentional) advantage, is that by having mutation occur after formation of the eDNA pool (see Figure 1), the eDNA pool in some sense acts like an "effective germ-line" that experiences a lower mutational load. This has the consequence of slightly lowering the effective mutation rate experienced by populations undergoing LGT, as is described below for a simplified case.

We recognise that this is a problem and have addressed it (see below).

We therefore request that the authors experiment with altering the ordering of events in their model (see Figure 1) to place the mutational step before the formation of the eDNA pool. This would allow the benefits arising from transformation to be easily distinguished from the benefits potentially arising from the smaller mutational load currently experienced by the eDNA pool. The outcome stands to range between two extremes:A) The ordering of these events has little effect on the qualitative results. The model is robust to these changes and the arguments of the paper are strengthened.B) It may turn out that the model behaviour observed so far is entirely dependent on the reduced mutational load of the eDNA pool (rather than transformation). Although it appears unlikely, this scenario would be difficult to justify from a biological perspective and would likely undermine the value of the paper.Issue of decreased mutational load of eDNA poolSuppose we begin with a population with no mutations. We use this ancestral population to (1) form the eDNA pool and (2) provide the samples for the new generation. Individuals in the new generation are then subject to mutation, receiving an average number of U mutations each. Let's assume for simplicity that the maximum number of mutations is 1. With probability λ the individual undergoes LGT, replacing part of its genome with a sample from the mutation-free eDNA pool. With the circular genome assumed, the probability of losing a mutation acquired by an individual in the new generation during LGT is L/g (recombination length / genome length). An effective mutation rate, U_eff_, in this simplified scenario can then be definedU_eff_ = U (1 – λ L /g)= \mu g ( 1 – λ L /g) [constant mutation rate per locus]This has a number of straightforward consequences.As g -> inf with L held constant, the effective mutation rate tends to the actual mutation rate. This could explain why the simulation results in Figure 2 with LGT and L=1-10 (which experience something like this effective mutation rate, U_eff_) tend to the result of the simulations without LGT (which experiences the actual mutation rate, U) as g gets large.As g -> inf with L held proportional to g (i.e. L=0.2g), the effective mutation rate is always strictly lower than the actual mutation rate. This could explain why the simulations in Figure 2 with L=0.2g always take longer to accumulate deleterious mutations than those without LGT.Finally, in Figure 2B the simulation results with LGT and L=10 are very close to those with L=0.2g when g is low (g\approx100). For these parameters, we have (L/g)=0.1 and (L/g)=0.2 respectively, which can lead to similar values of U_eff_, and thus could explain the observed similarity in the average extinction time of the LLC in this parameter range.The above discussion comes with the caveat that the numerical differences between U and this (admittedly crude) approximation for U_eff_ are very low for the parameters investigated (U_eff_ can be at most (1-λ)U=0.9U). Nevertheless it is important to understand which components of the model set-up are driving the dynamics. Placing mutation before the formation of the eDNA pool would circumvent the problem of these different effective mutation rates, and provide a fairer comparison of the robustness of populations under LGT and under no LGT to Muller's ratchet.

We thank the reviewers for the analysis they provide. We agree with the reviewers that having mutation occur after/before the formation of the eDNA pool can potentially affect the outcome of the model. We have therefore changed the order of these events, with mutation preceding the releases of DNA in the environment (Materials and methods, fourth paragraph) and repeated all the simulations using these new model dynamics.

Our new results are in agreement with the previous ones (slight quantitative differences are evident), and has little impact on our conclusions. We stick with the new ordering. The idea that the eDNA pool has a lower mutation load does not make a lot of sense and should be avoided. Thanks for pointing this out.